# C-C Motif Chemokine Ligand 2 Enhances Macrophage Chemotaxis, Osteogenesis, and Angiogenesis during the Inflammatory Phase of Bone Regeneration

**DOI:** 10.3390/biom13111665

**Published:** 2023-11-18

**Authors:** Issei Shinohara, Masanori Tsubosaka, Masakazu Toya, Max L. Lee, Junichi Kushioka, Masatoshi Murayama, Qi Gao, Xueping Li, Ning Zhang, Simon Kwoon-Ho Chow, Tomoyuki Matsumoto, Ryosuke Kuroda, Stuart B. Goodman

**Affiliations:** 1Department of Orthopaedic Surgery, Stanford University School of Medicine, Stanford, CA 94063, USA; issei27@stanford.edu (I.S.); masanori.tsubosaka@gmail.com (M.T.); mtoya@stanford.edu (M.T.); maxlee12@stanford.edu (M.L.L.); junichi5@stanford.edu (J.K.); muramasa@stanford.edu (M.M.); qigao7@stanford.edu (Q.G.); xuepli@stanford.edu (X.L.); skhchow@stanford.edu (S.K.-H.C.); 2Department of Orthopaedic Surgery, Kobe University Graduate School of Medicine, Kobe 650-0017, Japan; matsun@m4.dion.ne.jp (T.M.); kurodar@med.kobe-u.ac.jp (R.K.); 3Department of Orthopaedics and Traumatology, The Chinese University of Hong Kong, Hong Kong; ningzhang@cuhk.edu.hk; 4Department of Bioengineering, Stanford University School of Medicine, Stanford, CA 94305, USA

**Keywords:** angiogenesis, CCL2, inflammation, macrophage, migration, MSCs, osteogenesis

## Abstract

Local cell therapy has recently gained attention for the treatment of joint diseases and fractures. Mesenchymal stem cells (MSCs) are not only involved in osteogenesis and angiogenesis, but they also have immunomodulatory functions, such as inducing macrophage migration during bone regeneration via macrophage crosstalk. C-C motif chemokine ligand 2 (CCL2), a known inflammatory mediator, is associated with the migration of macrophages during inflammation. This study examined the utility of CCL2 as a therapeutic target for local cell therapy. Using lentiviral vectors for rabbit MSCs, genetically modified *CCL2* overexpressing MSCs were generated. Osteogenic differentiation assays were performed using MSCs with or without macrophages in co-culture, and cell migration assays were also performed. Additionally, co-cultures were performed with endothelial cells (ECs), and angiogenesis was evaluated using a tube formation assay. Overexpression of *CCL2* did not affect bone formation under monoculture conditions but promoted chemotaxis and osteogenesis when co-cultured with macrophages. Furthermore, *CCL2*-overexpression promoted tube formation in co-culture with ECs. These results suggest that CCL2 induces macrophage chemotaxis and osteogenesis by promoting crosstalk between MSCs and macrophages; CCL2 also stimulates ECs to induce angiogenesis. These findings indicate that CCL2 may be a useful therapeutic target for local cell therapy in areas of bone loss.

## 1. Introduction

Non-traumatic osteonecrosis of the femoral head (ONFH) is a refractory disease that typically causes progressive collapse of the femoral head and degenerative arthritis of the hip [1,2]. Local cell therapy has recently gained attention for the treatment of refractory joint diseases, such as ONFH [3]. A notable discovery in ONFH is the inhibition of differentiation of mesenchymal stem cells (MSCs) into osteoblasts [4]. Also, capillaries that act as conduits to deliver nutrients, MSCs, and other cells to the bone repair unit in ONFH can be occluded by thrombus or embolism [5]. Therefore, osteogenesis and angiogenesis are important for bone regeneration in ONFH, and MSCs may be a potential strategy for treatment. MSCs are not only crucial for bone formation by differentiating into osteoblasts and chondrocytes [6] but also possess immunomodulatory functions, including the induction of macrophage chemotaxis during bone regeneration [7,8]. MSCs also promote angiogenesis by interacting with endothelial cells (ECs). Successful bone regeneration is based on cooperative crosstalk between MSCs, macrophages, and ECs [8,9].

C-C motif chemokine ligand 2 (CCL2), a member of the CC chemokine superfamily [10], has immunomodulatory functions during inflammatory responses [11]. CCL2 function in the bone microenvironment has been reported to enhance the immunomodulatory function of MSCs and promote macrophage and EC migration during inflammation [11,12]. CCL2 has also been reported to act directly on EC receptors to induce angiogenesis [13] and to upregulate signaling pathways associated with increased VEGF production [14].

Considering these facts, we hypothesize that CCL2 is a mediator that regulates cellular crosstalk between MSCs, macrophages, and ECs, leading to improved osteogenesis and angiogenesis. In this study, employing a rabbit primary cell culture model, we investigated the therapeutic potential of CCL2-mediated cell therapy using genetically modified CCL2-releasing MSCs and recombinant CCL2 protein.

## 2. Materials and Methods

### 2.1. Rabbit MSCs

Rabbit bone marrow-derived MSCs purchased from Cyagen Biosciences (Santa Clara, CA, USA) were used for rabbit MSCs. MSCs were cultured in an α-minimal essential medium (α-MEM, Thermo Fisher Scientific, Waltham, MA, USA) supplemented with 10% certified fetal bovine serum (FBS, Thermo Fisher Scientific, Waltham, MA, USA) and 1% antibiotic/antifungal solution (A/A, Thermo Fisher Scientific). The following experiment used MSCs from passages 4 to 8.

### 2.2. Preparation of Rabbit CCL2 Plasmid

First, vectors containing elements of rabbit *CCL2* were produced according to a previous report [15]. Briefly, cytomegalovirus (CMV) promoter-driven constitutive rabbit *CCL2* expression lentivirus was released from the *CCL2*-expressing pCMV–r*CCL2*–His plasmid via digestion with the Spel/NotI restriction enzyme. Then, ligation to the pCDH–CMV copRFP lentiviral expression vector (CD511B-1; System BioSciences, Palo Alto, CA, USA) was performed to generate the pCDH CMV–r*CCL2*–copRFP vector.

### 2.3. Establishment of Genetically Modified Rabbit MSCs

In accordance with previous reports [11,15], human embryonic kidney 293T cells (ATCC, Manassas, VA, USA) were transfected with 25 M chloroquine using a calcium phosphate transfection kit (Clontech, Mountain View, CA, USA), psPAX2 packaging vector and pMD2G VSV-G envelope vector, and rabbit CCL2-secreted pCDH–CMV–r*CCL2*–copRFP-expressing lentiviral vectors. Empty vectors were also prepared using the lentiviral vector pCDH–CMV–copRFP (CD511B-1; System Biosciences, Palo Alto, CA, USA) as a control. After 2 days of incubation, the supernatant containing the virus was collected, mixed with 3 ug/mL polybrene in serum-free medium, and incubated at a multiplicity of infection (MOI) of 100 for 6 h to infect rabbit MSCs (Figure 1a). The virus infection was confirmed by RFP positivity using a fluorescence microscope (BZ-X810, KEYENCE, Osaka, Japan) 3 days after infection (Figure 1b).

### 2.4. Cell Cultures and Groups

Rabbit recombinant CCL2(rmCCL2) protein (Bio-Rad, Hercules, CA, USA) was used for temporary stimulation, and the dose was determined based on the manufacturer’s calibration curve; the absence of CCL2 cytotoxicity was confirmed using alamarBlue Cell Viability Reagent (Thermo Fisher Scientific), as in previous reports [16,17]. The following four groups were used for subsequent experiments with rabbit MSCs (Figure 2a): (1) MSCs without any intervention; (2) MSCs+rmCCL2: control MSCs without any genetic modification that were incubated with temporary rmCCL2 stimulation (single addition of 10 ng/mL at 24 h); (3) virus^+^MSCs: MSCs infected with an empty lentiviral vector; and (4) rCCL2^+^ MSCs: MSCs infected with rabbit CCL2-secreting lentiviral vectors.

### 2.5. Quantitative Real-Time Polymerase Chain Reaction (qPCR)

To confirm the success of the genetic modification, PCR was performed to evaluate the *CCL2* gene expression. Messenger RNA (mRNA) was extracted using Trizol reagent and reverse transcribed to cDNA using iScript Reverse Transcription Supermix for RT-qPCR (Bio-Rad, Hercules, CA, USA), followed by qPCR to quantify gene expression. Glyceraldehyde 3-phosphate dehydrogenase (*GAPDH*, forward: 5′-GCGCCTGGTCACCAGGGCTGCTT-3′; reverse: 5′-TGCCGAAGTGGTCGTGGATGACCT-3′) was used as the housekeeping gene; gene expression levels of *CCL2* (forward: 5′-GTCTCTGCAACGCTTCTGTGCC-3′; reverse: 5′-AGTCGTGTGTTCTTGGGTTGTGG-3′) were standardized and compared between MSCs and rCCL2^+^MSCs.

### 2.6. Migration Assay

To confirm the secretion and function of CCL2 from rCCL2^+^MSCs, a scratch assay was performed to evaluate macrophage cell migration ability based on previous reports [11,18]. Primary cells of rabbit bone marrow M0 macrophages (Cell biologist, Chicago, IL, USA) were used to perform indirect co-cultures with MSCs using transwell. MSCs were seeded on top of the transwells after seeding the macrophages (Appendix A). For macrophage culture, basic medium (RPMI 1640, 30% L929 leukocyte conditioned medium, 10% FBS, 1% A/A, 10 ng/mL macrophage colony-stimulating factor (M-CSF, R&D systems, Minneapolis, MN, USA)) was used. For co-culture, a 1:1 mixture of MSCs and macrophage basal medium in 500 μL was used. Cells were seeded into 24-well plates (2.0 × 10^4^ macrophages and MSCs, each) and cultured overnight, and the bottom of each well was scratched once using a 200 µL micropipette. After scratching, the progress of cell migration was observed using BZ-X810 (KEYENCE), and the cell blank areas and the distances between the areas were measured using QuPath (Version 0.4.3) [19]. The distance of cell blank areas was calculated as the average of the top, center, and bottom of each image taken, and the area of the blank areas was measured using QuPath. The results of each treatment were compared at 0, 6, 24, and 72 h after scratching among 5 groups: (1) macrophages alone, (2) temporary stimulation of macrophages with rmCCL2 protein (single addition of 10 ng/mL in 24 h) and indirect co-culture with (3) MSCs, (4) virus^+^MSCs, and (5) rCCL2^+^MSCs.

### 2.7. Cell Proliferation Assay

Cell proliferation was assessed using alamarBlue Cell Viability Reagent (Thermo Fisher Scientific) following the manufacturer’s protocol. In total, 1.0 × 10^4^ cells/100 μL from each group were seeded and cultured in 96-well plates, then 10 μL of alamarBlue reagent was added to each well and incubated at 37 °C for 1 h [17]. One hour later, absorbance was measured using SpectraMax iD3 (Molecular Devices, San Jose, CA, USA). The absorbance at 570 nm and 600 nm was measured, and the cell proliferation rate was calculated as in previous reports [11,16]. Following the manufacturer’s protocol, the average absorbance value at 600 nm of the cell culture medium was only subtracted from the absorbance value at 570 nm of the experimental wells and evaluated relative to the time course.

### 2.8. Osteogenic Differentiation Assay

Osteogenesis was assessed using alkaline phosphatase (ALP) and alizarin staining. Cells were seeded into 24-well plates overnight (ALP staining; 4.0 × 10^4^/500 μL, alizarin staining; 2.0 × 10^4^/500 μL) and incubated in osteogenic medium (α-MEM (Thermo Fisher Scientific), 10% fetal bovine serum (FBS, Thermo Fisher Scientific), 1% antibiotic/antifungal solution (A/A, Thermo Fisher Scientific), 10 mM β-glycerol phosphate (Sigma–Aldrich, St. Louis, MO, USA), 50 μM 1 ascorbic acid (Sigma–Aldrich), and 100 nM dexamethasone (Sigma–Aldrich)). Osteogenesis was evaluated in monoculture in four groups: (1) MSCs; (2) virus^+^ MSCs; (3) MSCs + rmCCL2; and (4) rCCL2^+^MSCs.

In addition, to investigate the effect of CCL2 on cell–cell interactions between MSCs and macrophages, direct co-cultures of MSCs and rabbit bone marrow M0 macrophages (Cell biologist, Chicago, IL, USA) were also performed for the following three groups: (1) MSCs; (2) MSCs+rmCCL2; and (3) rCCL2^+^MSCs. The co-culture medium was a mixture of osteogenic medium and basic medium with macrophages in a 1:1 ratio to 500 μL, and macrophages and MSCs were directly mixed and cultured at 2.0 × 10^4^ each for ALP staining and 1.0 × 10^4^ each for alizarin staining. ALP staining was performed with 1 StepTM NBT/BCIP substrate solution (Thermo Fisher Scientific) on day 7 of osteogenic induction, and bone matrix deposition was assessed with alizarin red staining (pH 4.1, Sigma–Aldrich) on day 21 of osteogenic induction [20]. Images were captured using a BZ-X810 (KEYENCE), and the percentage of ALP and alizarin staining was quantified using QuPath [19].

### 2.9. Tube Formation Assay

Twenty-four-well plates were bottom-coated with 200 μL of Matrigel (3.3 mg/mL; BD Life Sciences, Glendale, AZ, USA), as previously reported. [21,22]. Matrigel was polymerized at 37 °C for 2 h, and 2.0 × 10^4^ MSCs were seeded overnight and then co-cultured with 2.0 × 10^4^ rabbit ECs (Cell Biologist, Chicago, IL, USA). Basal medium (Cell biologist, Chicago, IL, USA) without growth factors [23] was used to culture ECs and was mixed 1:1 with an MSC basal medium for co-culture to 500 μL. After 72 h from incubation, tube formation was compared in the following four groups: (1) ECs alone and direct co-culture with (2) MSCs, (3) MSCs+CCL2, and (4) rCCL2^+^MSCs. Four microscopic fields were randomly selected and captured using BZ-X810 (KEYENCE). Tube formation was examined by counting the number of branches and the total vessel length using ImageJ plug-in software (version 1.8) [24]. 

### 2.10. Statistical Analysis

Statistical analyses were performed using Prism 9 (GraphPad Software, San Diego, CA, USA). All data are expressed as mean values ± standard deviations. The Mann–Whitney U test was used for comparisons between two groups, and one-way ANOVA with Tukey’s multiple comparison test was used for comparisons among multiple groups. Results with *p* < 0.05 were considered statistically significant.

## 3. Results

### 3.1. CCL2 Gene Expression

Successful genetic modification of MSCs was confirmed by mRNA expression by rt-PCR. Gene expression levels of *CCL2* were significantly higher in the rCCL2^+^MSCs group than in the MSCs group (*p* < 0.01), as shown in Figure 1c.

### 3.2. Cell Proliferation

Cell proliferation of MSCs was compared at 0, 24, 48, 72 h, and 1 week based on absorbance. Cell proliferation rates were not significantly different among the groups at each time point (Figure 2b). These results indicated that both viral transfection and exposure to rmCCL2 had no negative effect on MSC proliferation.

### 3.3. Osteogenic Differentiation (Monoculture)

MSCs monoculture in osteogenic differentiation medium showed no significant differences in positive area for ALP staining (MSCs: 29.9 ± 0.5%; MSCs + rmCCL2: 33.0 ± 3.0%; virus^+^MSCs: 27.7 ± 1.2%; rCCL2^+^MSCs: 31.6 ± 4.1%) and alizarin staining (MSCs: 37.6 ± 3.4%; MSCs + rmCCL2: 41.2 ± 7.8%; virus^+^MSCs: 44.0 ± 7.3%; rCCL2^+^MSCs: 44.3 ± 4.9%) between groups (Figure 3). As a result, lentiviral transgenesis and exposure to CCL2 in monoculture did not affect subsequent bone formation in vitro.

### 3.4. Migration Assay

Migration assay results using the scratch test showed that residual scratch marks in macrophages after 72 h disappeared in the rCCL2^+^MSCs group (Figure 4a). The ratio of the distance between macrophages at 0 and 24 h after scratching (24 h/0 h) was significantly reduced in the rCCL2^+^MSCs group compared to macrophages alone, MSCs group, and virus+MSCs group (Figure 4b, macrophages alone: 0.73 ± 0.07; macrophages with rmCCL2: 0.65 ± 0.05; MSCs: 0.73 ± 0.03; virus^+^MSCs: 0.79 ± 0.09; rCCL2^+^MSCs: 0.53 ± 0.06, *p* < 0.05). The area ratio showed a significant reduction in macrophages with rmCCL2 and rCCL2^+^MSCs groups compared to the other three groups (Figure 4c, macrophages alone: 0.77 ± 0.08; macrophages with rmCCL2: 0.51 ± 0.02; MSCs: 0.71 ± 0.04; virus^+^MSCs: 0.75 ± 0.04; rCCL2^+^MSCs: 0.47 ± 0.09, *p* < 0.01). These results suggest that recombinant CCL2 promotes macrophage migration and that secretion of CCL2 from rCCL2^+^ MSCs is also a function.

Since the lentivirus itself was shown to have no effect on cell proliferation, macrophage migration, and osteogenesis, the virus^+^MSCs group was excluded from the following experiments.

### 3.5. Osteogenic Differentiation (Co-Culture)

When MSCs were co-cultured with macrophages, the positive area in ALP staining was significantly higher in the groups temporarily stimulated with rmCCL2 and in the rCCL2^+^MSCs group than in the MSCs group (Figure 5, MSCs: 16.5 ± 3.8%; MSCs + rmCCL2: 33.5 ± 3.2%; rCCL2^+^MSCs: 33.6 ± 9.0%; *p* < 0.01). The positive area in alizarin red staining was also significantly higher in the groups temporarily stimulated with rmCCL2 and in the rCCL2^+^MSCs group than in the MSCs group (Figure 5, MSCs: 24.4 ± 3.0%; MSCs + rmCCL2: 40.0 ± 7.9%; rCCL2^+^MSCs: 37.9 ± 4.3%; *p* < 0.05).

### 3.6. Angiogenesis (Tube Formation Assay)

After 72 h of incubation of MSCs and ECs without growth factors, tube formation was observed in the MSCs+rmCCL2 and rCCL2^+^MSCs groups (Figure 6a,b). Quantification using ImageJ showed a significant increase in the number of branches when co-cultured with the three groups containing MSCs compared to ECs alone (Figure 6c, ECs alone: 8.3 ± 2.6; MSCs: 38.8 ± 9.1; MSCs + rmCCL2: 39.8 ± 10.3; rCCL2^+^MSCs: 40.5 ± 4.8; *p* < 0.05). Branch elongation and segment formation were not sufficient for the co-culture of ECs and MSCs alone but were significantly increased by exposure to CCL2. As a result, total vessel length was significantly increased by rmCCL2 stimulation and secretion of CCL2 from rCCL2^+^MSCs (Figure 6c, ECs alone: 2.1 ± 0.2 mm; MSCs: 2.3 ± 0.7 mm; MSCs + rmCCL2: 6.3 ± 1.7 mm; rCCL2^+^MSCs: 6.2 ± 1.2 mm; *p* < 0.01).

## 4. Discussion

In this study, we used genetically modified CCL2-releasing MSCs to verify the hypothesis that CCL2 is a mediator that regulates cellular crosstalk among MSCs, macrophages, and ECs, leading to improved osteogenesis and angiogenesis. CCL2 did not affect osteogenesis under monoculture conditions of MSCs but promoted migration capacity and osteogenesis when co-cultured with macrophages. Also, when co-cultured with endothelial cells, CCL2 promoted tube formation. These results suggest the following three functions of CCL2: CCL2 induces macrophage chemotaxis; CCL2 promotes crosstalk between MSCs and macrophages, leading to enhanced osteogenesis; and CCL2 promotes crosstalk between MSCs and ECs, leading to enhanced angiogenesis. Thus, CCL2 may be a biological target to promote osteogenesis and angiogenesis in the inflammatory phase of bone regeneration.

CCL2 is an inflammatory chemokine that regulates leukocyte mobilization during inflammation and is also called monocyte chemotactic protein-1 [25]. CCL2 is known to bind to C-C motif chemokine receptor 2 (CCR2) to induce chemotactic activity and has been proposed as a potential therapeutic target for various human diseases [26]. In orthopedics, CCL2 has been reported to be associated with various disorders, including osteoarthritis and the regulation of bone formation [27,28]. With respect to the latter, genetic variants of CCL2 and CCR2 have been identified as risk factors for osteopenia [29], and CCL2 has been reported as an important regulator molecule in the skeletal system [30]. CCL2 is expressed by osteoblasts following an inflammatory response and is known to be involved in bone formation by stimulating monocyte migration and cell proliferation [31]. Our previous study in mice suggested that CCL2 has no direct effect on MSCs alone. We also showed that acute stimulation with CCL2 (single dose at 24 h) promoted macrophage chemotaxis and subsequent osteogenesis [11]. Furthermore, genetically modified CCL2 secretion and continuous stimulation with CCL2 suppressed bone formation, which underlined the important role of CCL2 at specific time points during the inflammatory phase [11]. The authors observed that genetically modified CCL2 secretion is similar to continuous CCL2 administration and may lead to bone loss due to prolonged inflammation. In this study, rabbits with low baseline CCL2 expression and possibly a high sensitivity to CCL2 were used [32], and the rCCL2^+^MSCs group also showed high osteogenic differentiation potential. This fact may explain the observations on why both temporary stimulation with CCL2 and secretion of CCL2 via rCCL2^+^MSCs promoted macrophage chemotaxis and osteogenesis, as reflected by ALP and alizarin staining. At the same time, it suggests that the concentration of CCL2 in bone differentiation is important and may vary among species.

CCL2 also regulates immune cells, including macrophages. The balance of polarity between inflammatory M1 macrophages (classically activated macrophages) and anti-inflammatory M2 macrophages (alternatively activated macrophages) is important in bone regeneration [33], and CCL2 is implicated in its regulation [34]. Decreased CCL2 results in increased polarization to M2 macrophages and decreased polarization to M1 macrophages [35]. Overexpression of CCL2 may prolong inflammation [36] and the balance between M1 and M2 macrophages. CCL2 is also involved in the migratory ability of macrophages. Knockdown of CCL2 and CCR2 in adipocytes impairs macrophage migration [37]. Crosstalk between MSCs and macrophages is important for bone regeneration, and CCL2 is key in enhancing macrophage chemotaxis. As in previous studies [11], macrophage chemotaxis was also promoted by CCL2 in this study. We were then able to infer that the promotion of osteogenesis by CCL2 was not the result of its direct effect on MSCs but indirectly through the promotion of macrophage migration and crosstalk between macrophages and MSCs. On the other hand, continued CCL2 stimulation may increase inflammatory M1 macrophages due to prolonged inflammation, leading to bone loss [11], suggesting that further investigation of CCL2 and macrophage polarization is needed.

Crosstalk between MSCs and ECs and the effects of CCL2 are important to angiogenesis [9]. MSCs and ECs interact synergistically through direct and indirect pathways in bone regeneration to promote angiogenesis and neovascularization [38]. Xu et al. reported that the co-culture of MSCs and ECs promotes the branching of new vessels [39]. ECs also promote osteogenesis in addition to angiogenesis when co-cultured with MSCs and are considered osteoinductive mediators during differentiation [9]. CCL2 also stimulates EC branching as well as tube formation [40,41,42]. CCL2 binds to CCR2 on ECs, resulting in the upregulation of transcription factors and stimulating signaling pathways, such as ERK1/2, PI3K, and MAPK, leading to EC proliferation and angiogenesis [14,43]. With respect to angiogenesis in this study, the co-culture of MSCs and ECs increased the number of vascular branches, and exposure to CCL2 promoted tube formation. These results are consistent with previous reports and suggest that CCL2 may be important in the interaction between MSCs and ECs.

This study has some limitations. First, this study was conducted in vitro. Further studies are needed to translate and validate these findings in vivo. Second, the studies are short-term; longer-term effects in vitro and in vivo need exploration. Third, further studies, including more cell types relevant to bone healing, such as osteoclasts and the polarity of macrophages, are needed to clarify the role of CCL2 in osteogenesis and bone remodeling, processes that are important for bone homeostasis. Fourth, changes in cell character and cell proliferation rate due to passaging or co-culturing of MSCs were not examined. Finally, we eliminated the continuous CCL2 group in this study since CCL2 secretion from genetically modified MSCs was not significantly different from continuous CCL2 administration reported in a previous study [11]. Further studies, including protein quantification, are needed to determine the ideal CCL2 concentration for osteogenic differentiation in vitro.

## 5. Conclusions

This in vitro study examined the feasibility of a locally delivered cell therapy of CCL2 using genetically modified MSCs versus recombinant protein. Both recombinant CCL2 and CCL2 secreted by genetically modified MSCs promoted macrophage chemotaxis during inflammation. Subsequent osteogenesis was then promoted when MSCs and macrophages were co-cultured in the environment of CCL2 exposure. Regarding angiogenesis, which is important for bone regeneration, exposure to CCL2 in the co-culture of MSCs and ECs also promoted tube formation. Local delivery of CCL2 during the acute phase of inflammation may be a strategy for local cell therapy in refractory cases of bone loss and diseases such as ONFH because it promotes chemotaxis of macrophages and MSCs and crosstalk among cells to enhance angiogenesis and osteogenesis.

## Figures and Tables

**Figure 1 biomolecules-13-01665-f001:**
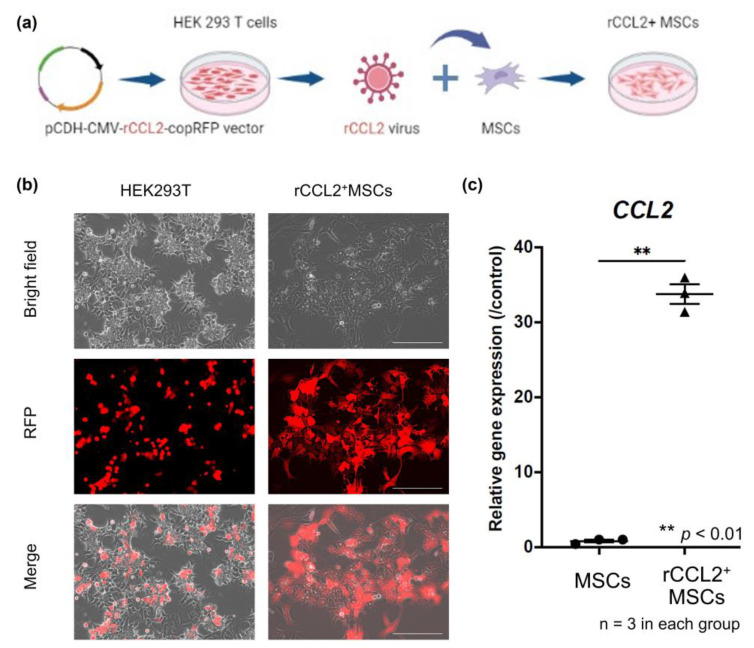
(**a**) Genetically modified schema for *CCL2* over-expressing MSCs. Rabbit C-C motif chemokine ligand 2 (rCCL2) secreted pCDH–CMV–m*CCL2*–copRFP expressing lentiviral vector was co-transfected with human embryonic kidney 293T cells. The supernatant containing the virus was diluted with mesenchymal stem cells (MSCs) culture medium to infect the MSCs. These MSCs infected with rCCL2-secreting lentiviral vectors were designated rCCL2^+^MSCs. (**b**) Infected cells with the virus were confirmed positive for RFP using fluorescence microscopy (scale bar = 100 μm). (**c**) Gene expression of *CCL2* by quantitative real-time polymerase chain reaction was significantly higher in the rCCL2^+^MSCs group than in the MSCs group (** *p* < 0.01, n = 3 in each group).

**Figure 2 biomolecules-13-01665-f002:**
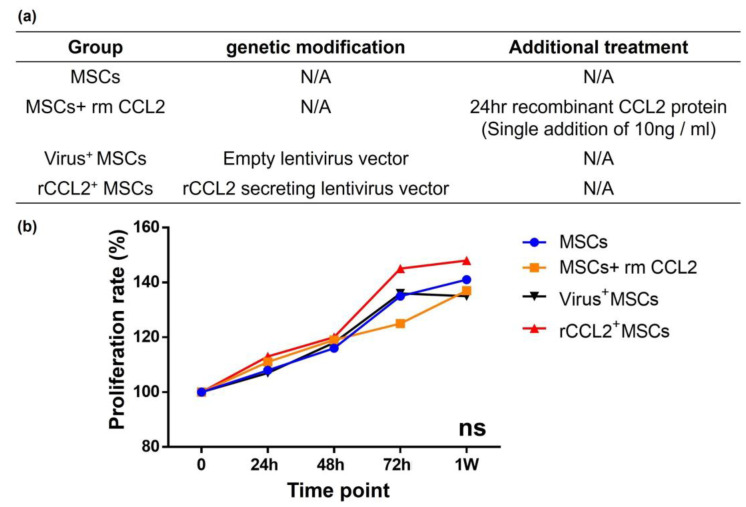
(**a**) Cell proliferation in different groups with/without genetic modification and therapeutic intervention. MSCs: mesenchymal stem cells; rm: recombinant; CCL2: C-C motif chemokine ligand 2; rCCL2: rabbit CCL2; (**b**) there was no difference in cell proliferation rate among the four groups.

**Figure 3 biomolecules-13-01665-f003:**
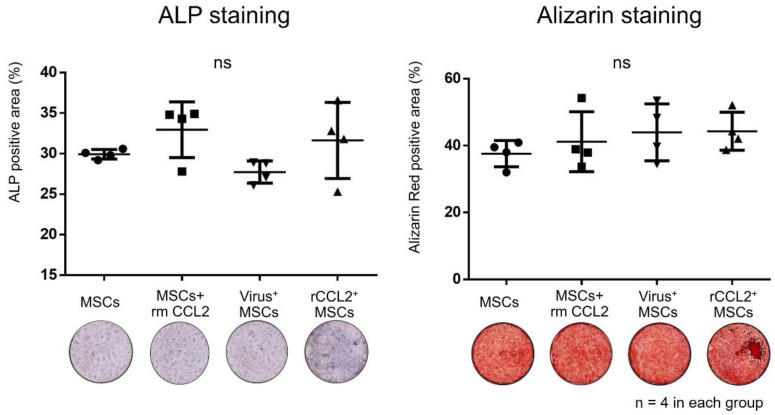
The results of the quantitative analysis of the percentage of alkaline phosphatase (ALP) and alizarin red positive area (%/well) in each group were not significantly different (ns, n = 4 in each group).

**Figure 4 biomolecules-13-01665-f004:**
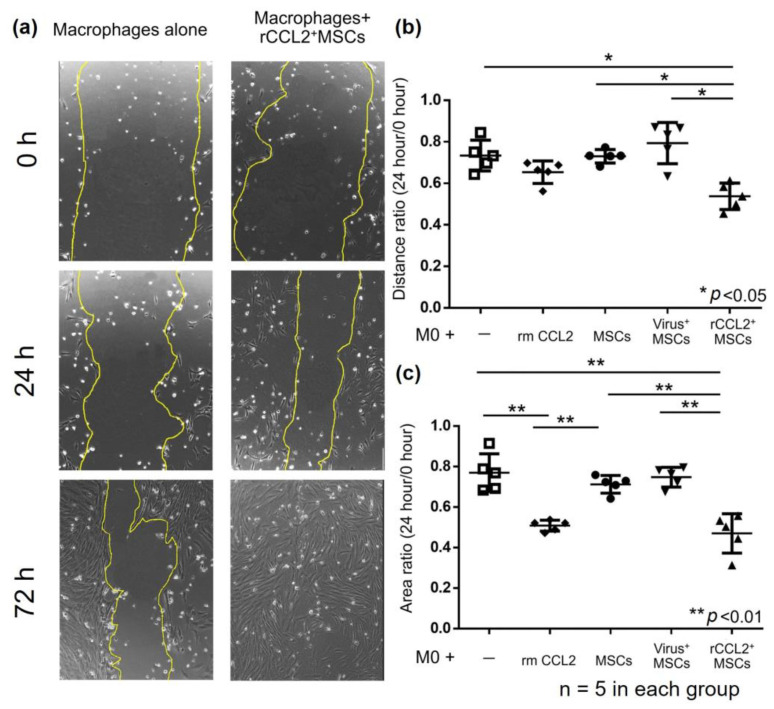
(**a**) Representative images of the scratch test in the following groups: macrophages alone and macrophage indirect co-culture with rCCL2^+^MSCs at 0, 24, and 72 h after scratching. Scratch marks disappeared in the rCCL2^+^MSCs group after 72 h. (**b**) Migration assay results showed that the rCCL2^+^MSCs group significantly reduced the distance ratio (24 h/0 h) compared with macrophages alone, MSCs, and virus^+^MSCs groups (* *p* < 0.05, n = 5 in each group). (**c**) The area ratio was significantly reduced in the macrophage with recombinant CCL2 group and the rCCL2^+^MSCs group compared to the other three groups (** *p* < 0.01, n = 5 in each group).

**Figure 5 biomolecules-13-01665-f005:**
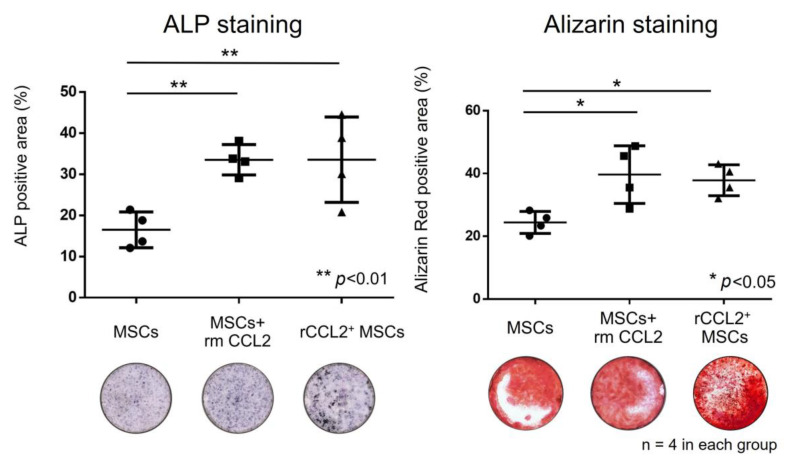
In co-culture of MSCs and macrophages, the percentage of the area positive for ALP (%/well) increased significantly in the rmCCL2 group and the rCCL2^+^MSCs group (** *p* < 0.01, n = 4 in each group), and the percentage of the area positive for alizarin red (%/well) increased significantly in the rmCCL2 group (* *p* < 0.05, n = 4 in each group).

**Figure 6 biomolecules-13-01665-f006:**
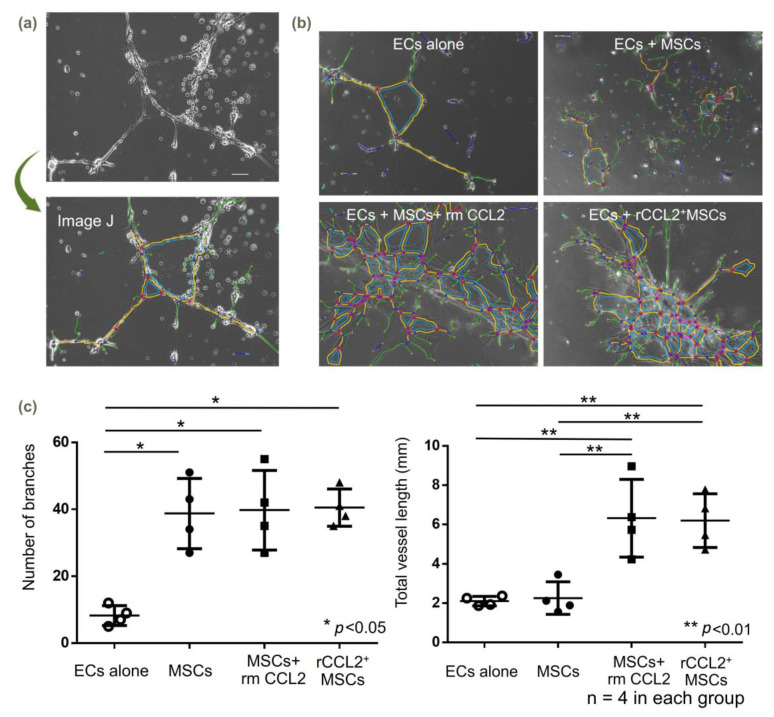
Tube formation assay. (**a**) Quantification method of vessel branching using Image J (version 1.8) and (**b**) representative images. The colored lines are generated automatically by ImageJ. ECs: endothelial cells. (**c**) In the tube formation assay, the number of branches was significantly increased co-culture with MSCs with and without CCL2 addition or over-expression (* *p* < 0.05, n = 4 in each group), and total vessel length was significantly increased with CCL2 exposure (** *p* < 0.01, n = 4 in each group).

## Data Availability

The data presented in this study are available on request from the corresponding authors. The data are not publicly available due to confidentiality concerns.

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
