# Peer review of "C-C Motif Chemokine Ligand 2 Enhances Macrophage Chemotaxis, Osteogenesis, and Angiogenesis during the Inflammatory Phase of Bone Regeneration"

_biomolecules, 2023, doi:10.3390/biom13111665_

Round 1

Reviewer 1 Report

Comments and Suggestions for Authors

Authors observed increases in osteogenic differentiation, chemotaxis, and angiogenesis when co-culturing macrophages with rabbit MSCs overexpressing CCL2 or adding rabbit recombinant CCL2 (rmCCL2) into the co-culture. Such effects were not observed in monoculture MSCs in the absence of macrophages. Based on the in vitro results, the authors conclude that CCL2 promotes crosstalk between macrophages and MSCs, and CCL2 could be useful for local cell therapy for bone defects. My comments /concerns and suggestions are listed below:

(1)   Passages 4 to 8 rabbit MSCs were used in the experiments. It is reported that MSC passage can significantly affect the differentiation capability of the cells.   It should be clarified whether the osteogenic differentiation potential of passages 4 to 8 of rabbit MSCs used for the study remained constant.

(2)   The description of migration assay (page 3, lines 105 to 121) is somewhat confusing and difficult to understand. What is the indirect co-culture? A diagram to illustrate the assay would be helpful.  

(3)   Please specify the amount (number) of cells (MSCs, macrophages, and ECs) used for the assays reported in sections 2.6, 2.7, 2.8, and 2.9.

(4)   Sections 2.7, 2.8, and 2.9: please specify the total volume of the medium in each well used for the assays. 

(5)   Section 2.8, what was the direct co-culture? Did it mean directly mixing macrophages and MSCs? The description from lines 137 to 146 on page 4 is somewhat confusing; please revise.

(6)   Figure 2. The cell proliferation assay was performed for only three days (72 hours). Longer duration should be done to generate more convincing data. The MTT assay may be used to confirm the result. There were four groups, but the legend indicates five groups. Please explain how to acquire (or calculate) the proliferation rate in the alamarBlue assay.   

(7)   Page 6, lines 195-196: No in vivo experiment was performed. Based on only in vitro osteogenic differentiation assay,  it is not readily concluded that ” As a result, lentiviral transgenesis and exposure to CCL2 in monoculture did not affect subsequent bone formation.”

(8)   Figure 4: Why the rmCCL2 treatment group was not included in this assay? It looks like n=5 in Figure 4b, but on page 7, line 215 indicates n=4, please clarify. Besides the cell scratch assay, confirming the results with other chemotaxis assays, such as the Boyden Chamber Assay, would strengthen the conclusion. 

(9)   Figure 5: The images of Alizarin staining show minimal differences among the treatments and control.   More quantitative measurements (such as calcium quantification) should be done to verify the result. 

(10) Figure 6: the corresponding imageJ micrographs of Figure 6b should be provided—no typic branches and vessels in the treatment groups of ECs+MSCs+rmCCL2 and ECs+rCCL2+MSCs in Figure 6b.   

Comments on the Quality of English Language

English is acceptable.  Some minor revisions are needed:

page 3, lines 105 to 121.

Page 4, lines 137 to 146. 

Author Response

Thank you very much for your comments that have helped us to improve the quality of our manuscript. To the best of our abilities, we have modified the manuscript to incorporate your critical suggestions. We did not change the basic format or the conclusions of the previously submitted paper. Please find our point-by-point responses listed below. The changes in the manuscript are highlighted in yellow. Please see attached file.

Reviewer 2 Report

Comments and Suggestions for Authors

Previously, the authors prepared CCL2-overexpression mesenchymal stromal cells (MSCs) derived from mice. The co-culture with CCL2-over expression MCSs and macrophage did not enhance the calcified nodule formation evaluated alizarine red staining, whereas macrophage migration was induced in that co-culture. Thus, there is a difference between the present study using MSCs derived from rabbits and the previous study using MSCs derived from mice in the effects of CCL2-overexpression MSCs on osteogenesis. Although, the authors mentioned this difference in the part of discussion, some questions remain.

1.        Was the supplement of CCL2 by genetic CCL2 overexpression MSCs derived from rabbit  temporal or continuous?

2.        How do lower baseline CCL2 and higher sensitivity to CCL2 in rabbits have the same effect as temporary CCL2 supplementation against calcified nodule formation and differentiation to osteoblasts in co-culture of MSCs and macrophage?

3.        Do continuous supplement of CCL2 have no effect on calcified nodule formation in co-culture of mice MSCs and macrophage?

4.        Did macrophage produce any kind of cytokine that induce differentiation of MCSs into osteoclast-like cells when co-culturing with MSCs that expressed CCL2?   

Author Response

We wish to express our appreciation to your insightful comments, which have helped us significantly improve the paper. The changes in the manuscript are highlighted in yellow. Please see attached file.

Reviewer 3 Report

Comments and Suggestions for Authors

The article by Shinohara et al., demonstrates the effect of CCL2 on macrophage chemotaxis, osteogenesis, and angiogenesis during the inflammatory phase of bone regeneration. This is a critical study as regenerative medicine suign stem cells gaining momentum. The article is well thought out, designed, and performed. The results support the hypothesis. The authors need to address a few comments and concerns before the manuscript moves to the next stage.

As the monoculture between the groups did not show any difference in cell proliferation rate or in functional analysis, the coculture with CCL2 and macrophages or with EC cells showed significant functional differences. This is interesting. However, the effect of coculture on the cell proliferation rate is now shown while it is shown for mycoculture – Figure 2.

The authors should show the effect of coculture on cell proliferation rate- hopefully, this can be easily achieved by FACS separation as the MSCs are tagged with RFP protein. This is important to show whether CCL2 is important in driving functional phenotype or also proliferation.

Will recombinant CCL2 protein replicate the migratory phenotype of macrophages?

Minor comments

There are a few typos

1.       Fig. 1C: why is there a p-value at the bottom and the ** at the top representing significance?

2.       Fig. 2 The legends describe five groups while the graph shows only four groups; need to correct this.

Comments on the Quality of English Language

No comments 

Author Response

Thank you very much for reviewing and commenting on our manuscript. We have responded to each of your comments and revised the manuscript accordingly. Please see attached file.

Round 2

Reviewer 1 Report

Comments and Suggestions for Authors

The revision has addressed most of my comments. I have the following two comments:

(a)    Osteogenic differentiation assay, Page 4: Please specify how many days of osteogenic induction before ALP staining and Alizarin staining.

(b)    Migration assay: 2.6; 3.4. For this type of assay, the scratches should be created when cells are grown to high confluence, forming a monolayer. However, based on Figure 4a, the scratch was created at very low cell confluence (shown in 0 hours). It is my concern that the observed effects were due to the proliferation of the macrophages at the low cell density; i.e., rCCL2+MSCs promoted the proliferation of macrophages, resulting in disappearing residual scratch marks after 72 hours.    

Author Response

We wish to express our appreciation to your insightful comments, which have helped us significantly improve the paper. Please see attached file.

Reviewer 3 Report

Comments and Suggestions for Authors

Thanks to the authors for promptly addressing the comments 

Author Response

We would like to express our great appreciation to you for the valuable comments and suggestions on our manuscript.
